# A Novel ROI Extraction Method Based on the Characteristics of the Original Finger Vein Image

**DOI:** 10.3390/s21134402

**Published:** 2021-06-27

**Authors:** Huimin Lu, Yifan Wang, Ruoran Gao, Chengcheng Zhao, Yang Li

**Affiliations:** School of Computer Science and Engineering, Changchun University of Technology, Changchun 130102, China; 2202003035@stu.ccut.edu.cn (Y.W.); 2202003047@stu.ccut.edu.cn (R.G.); 2202003113@stu.ccut.edu.cn (C.Z.); liyang1979@ccut.edu.cn (Y.L.)

**Keywords:** biometrics, finger vein recognition, ROI extraction, identity authentication

## Abstract

As the second generation of biometric technology, finger vein recognition has become a research hotspot due to its advantages such as high security, and living body recognition. In recent years, the global pandemic has promoted the development of contactless identification. However, the unconstrained finger vein acquisition process will introduce more uneven illumination, finger image deformation, and some other factors that may affect the recognition, so it puts forward higher requirements for the acquisition speed, accuracy and other performance. Considering the universal, obvious, and stable characteristics of the original finger vein imaging, we proposed a new Region Of Interest (ROI) extraction method based on the characteristics of finger vein image, which contains three innovative elements: a horizontal Sobel operator with additional weights; an edge detection method based on finger contour imaging characteristics; a gradient detection operator based on large receptive field. The proposed methods were evaluated and compared with some representative methods by using four different public datasets of finger veins. The experimental results show that, compared with the existing representative methods, our proposed ROI extraction method is 1/10th of the processing time of the threshold-based methods, and it is similar to the time spent for coarse extraction in the mask-based methods. The ROI extraction results show that the proposed method has better robustness for different quality images. Moreover, the results of recognition matching experiments on different datasets indicate that our method achieves the best Equal Error Rate (EER) of 0.67% without the refinement of feature extraction parameters, and all the EERs are significantly lower than those of the representative methods.

## 1. Introduction

In the field of Biometrics, security and efficiency are vital for recognition systems. Moreover, the recognition accuracy and recognition rate also need to be taken into account. In 2000, Kono et al. [1] proposed a method to use finger vein patterns for authentication. Finger vein characteristics have very obvious advantages over other biophysiological characteristics such as: (1) vivo identification: finger vein information can only be obtained on living human beings; (2) uniqueness: T. Yanagawa et al. [2] verified the uniqueness of each finger vein pattern; (3) stability: the finger vein distribution characteristics of each individual remain the same throughout life after adulthood; (4) security: finger veins are internal features of the human body, and are difficult to falsify due to their high complexity. Therefore, finger vein recognition has attracted a great deal of attention from academia and enterprises with its series of advantages, and has made great progress in recent years [3].

Finger vein recognition generally includes four main steps: image acquisition, preprocessing, feature extraction, and matching recognition. In the process of finger vein image acquisition, the image quality is relatively poor because it is easily affected by environmental factors such as uneven illumination and temperature, and also by personal factors such as finger image translation, rotation, scaling, and gesture changes. The global pandemic in recent years has raised public health safety issues to an unprecedented level, prompting the development of acquisition methods in a contactless, unconstrained direction, which has exacerbated the impact of external factors on acquisition effectiveness. Extracting the vein information from the fixed area of a finger as Region of Interest (ROI) for feature extraction, ensures the minimal differences between the features obtained for identification of the same individual under different conditions [4]. Therefore, a robust, accurate, and fast ROI extraction method is crucial in preprocessing, which largely determines the efficiency and reliability of the finger vein recognition system. The goal of finger vein ROI extraction is to maximize the retention of valuable regions and remove useless regions from the acquired images, which requires addressing three aspects: finger region segmentation, image correction, and vertical reference line positioning. At present, researchers have proposed many solutions to these three problems, but the existing methods are not very satisfactory in terms of reliability, robustness, and efficiency due to various external factors, such as image translation and rotation, as well as uneven illumination or background interference. Therefore, we endeavor to explore a robust and efficient ROI extraction method for finger vein images. In this paper, we propose a novel ROI extraction method based on the original finger vein image characteristics, which mainly includes:(1)A horizontal Sobel operator with additional weights is proposed to provide an accurate starting point for the search.(2)Observing the finger cross-sectional MRI image, we found that the finger cross-sectional surface showed a sub elliptical shape, so we propose a new finger edge detection method by using the imaging property of the finger contour in the vein image.(3)A large receptive field gradient operator and a new method for localizing joint cavities based on this operator are proposed, which solves the difficult problem of localizing joint cavities caused by large gray gradients in vein images.

The paper is organized as follows: Section 2 reviews the representative research results on ROI localization; Section 3 elaborates the proposed ROI extraction method for finger vein images and its innovative methods in each step; Section 4 designs and performs extensive experiments based on four different public datasets to test and validate the proposed method in terms of extraction time and recognition performance; Section 5 clarifies the research conclusions and gives an outlook for future research directions in this field.

## 2. Related Works

In recent years, researchers have devoted themselves to the study of ROI extraction of finger vein images. Generally, ROI extraction is performed in the process of preprocessing, which can be divided into ROI localization and extraction, and vein image enhancement and normalization. The accurate extraction of ROI is a prerequisite for image enhancement and normalization. According to the different ways of ROI localization, ROI extraction methods are divided into four categories as follows [5]:

The first class of methods is based on fixed window localization. Wang et al. [6] computed the binarized finger image center of mass and used the finger contour maximum external matrix as ROI; Rosdi et al. [7] used OTSU to binarize the finger vein image, then used the center of the finger region in the binarized image as the centroid, and selected a fixed window of size 480 × 160 as the ROI; Yang and Shi [8] first used a predefined fixed size window to remove part of the background, and then defined a fixed size window as the ROI in combination with the distal finger joint positioning. The predefined window size of this kind of methods are usually fixed, which determines that the methods cannot adapt to various changes in the image and are less robust.

The second class of methods is threshold-based localization. Kumar et al. [9] used 230 fixed threshold to obtain the binary image, then subtracted the binary edge image which was obtained by using Sobel operator. Furthermore, they analyzed the connected domain and removed the non-finger region part to obtain the finger region mask. Then, to obtain the ROI, S. Brindha [10] binarized the image and performed open and close operations on the obtained binarized image to obtain the complete finger region mask. Ideally, the pixel values in the finger area are all higher or lower than the background area. However, in practice, due to the different light conditions of different collection devices, it is difficult to avoid the distribution of pixel values in the local area of the finger having similarities with the background area, so these threshold-based methods are not very robust.

The third class of methods is mask-based localization. Lee et al. [11] divided the image into upper and lower parts, and designed two horizontal edge detection operators to convolve the image separately, then selected the maximum response value as the finger edge; Lu et al. [12], inspired by the literature [11], expanded the horizontal Pweett operator and then corrected the wrong edge using the directional angle to obtain the complete finger edge; Wang and Tang [13] used the operator proposed in the literature [11] and the Sobel operator for edge detection, and analyzed the connected domain on the obtained edges to remove line segments with lengths less than a fixed threshold; Song et al. [14] used the Laplace operator to detect edges and used the mean curvature to complement the finger edges. Many edge detection operators are very sensitive to noise, and the background noise distribution of images in different acquisition devices are all very different, making it easy to take noise as the maximum response and thus detect the wrong edge.

The fourth class of methods is based on fine-tuning finger region localization. Yang et al. [15] used sliding windows to localize the finger joint cavity; Wang et al. [13] further compared different shapes of sliding windows and concluded that elliptical windows are more suitable for finger joint cavity localization; Qiu et al. [16] used dual sliding windows which is more stable and accurate to localize the finger joint cavity; Yang et al. [17] combined super pixels for finger edge detection; Yao et al. [5] introduced the Kirsch edge detector to finger vein ROI localization for the first time, and set a three-level dynamic threshold in combination with the 3σ principle to obtain a complete finger edge. Such methods acquire a more complete ROI by further processing the extracted edges, but the customized processing is not robust and inefficient.

In order to develop a highly accurate authentication system, vein features need to be accurately extracted from the captured images, and this process must be performed quickly. Unfortunately, the existing methods cannot be robust, efficient, and accurate at the same time for image processing affected by complex background, scattering, and uneven illumination. Through a large number of studies, we found that the finger cross-sections are nearly elliptical, which generally produce a stable and obvious imaging property of the finger contour. We can use this feature to segment the finger region. In order to solve the difficult localization problem caused by the large gradient of the joint cavity, we propose a joint cavity localization method using the gradient operator of large receptive field to further divide the finger region. This proposed method including: (1) A new method which combines vein image with horizontal edge detection operator can detect finger edge efficiently and accurately; (2) A new large receptive field joint cavity inspection operator based on human visual characteristics; (3) Accurate search of joint cavity based on the proposed operator, and the accurate extraction of the vein image ROI.

## 3. The Proposed Method

An efficient and robust finger vein region extraction method is the key to finger vein recognition system. In this section, we elaborate the proposed method: A novel ROI extraction method based on the characteristics of the original finger vein image, which includes four steps: finger region segmentation, orientation correction, joint cavity localization, and ROI extraction. Research shows that the cross-sectional shape of the finger is approximately elliptical, and the edge of the finger is relatively thin, which is the brightest region in the whole image when capturing. We propose a finger edge search method based on the finger contour imaging characteristics. This method combines the original vein image with the improved edge detection operator to achieve accurate detection of finger edges in the local area, and obtains continuous single-pixel finger edge point coordinates. Due to the continuous finger edge, points do not occur with abrupt angle changes, and the search range of the proposed method can be limited to the upper and lower neighborhoods of the finger edges, which is less computationally intensive. After the finger edge coordinates are obtained, the least square estimation is used to calculate the correction angle, and the finger image is corrected by affine transformation.

Through the analysis, we found that the imaging of joint cavity often produces a large range of gray gradient, which causes the problem of difficult localization. Therefore, in this paper, we propose a gradient detection operator with a large receptive field, which can be used to achieve an accurate search of the distal joint cavity. The proposed method is based on the universal properties of finger vein imaging which are distinct and stable, and the accurate extraction of ROIs can be achieved using such properties. In addition, in order to improve the recognition efficiency, the original images are reduced to half of the original size by using “double triple” interpolation. Figure 1 illustrates the entire process of the ROI extraction method proposed in this paper, with each step was described in detail below.

### 3.1. Segmentation of Finger Region

In practical applications, we found that the threshold and mask methods are not very satisfactory in terms of processing time and segmentation accuracy due to the complex background of the acquired images. Many elements in the image acquisition process can cause incorrect finger region segmentation, such as unclear boundaries due to uneven illumination (Figure 2a), incorrect response of edge gradients due to background (Figure 2b), complex background pixel gray value distribution affecting the boundaries extracted by the thresholding method (Figure 2c), and disappearance of upper or lower boundaries due to finger displacement (Figure 2d). The above situations are common in the existing public datasets and practical applications. However, none of the existing methods can segment the finger region perfectly, so we propose a new finger edge search method based on the finger contour imaging characteristics to solve this problem.

#### 3.1.1. Improved Edge Detection Operator

The rules (for details see Section 3.1.2) first need to find the starting point of the search on the top and bottom edges of the finger, separately, so we need a simple and efficient method. The Sobel operator has better results in dealing with grayscale gradient type edges, but due to the presence of a large amount of noise in finger vein images, it has an impact on the localization of the starting point search. The traditional horizontal Sobel operator has the same weight for the gradients at each position, as shown in Equation (Equation 1).
(1)$$\begin{array}{cc} \; & G_{y}=\left[f\left(x-1,y+1\right)+2\times f\left(x,y-1\right)+f\left(x+1,y+1\right)\right]\\ \; & -[f(x-1,y+1)+2\times f(x,y+1)+f(x+1,y+1)] \end{array}$$

For finger vein images, it can be considered that the gradient closer to the center of the image is more likely to be the finger region. Meanwhile, the vein pattern inside the finger generates a gradient when the gradient is calculated for the horizontal direction. Therefore, a weight that has a fast rate of change in the background region and a slow rate of change in the internal finger region is needed. To achieve accurate localization of the search starting point, we propose an improved horizontal Sobel operator, as shown in Equation (Equation 2).
(2)$${G_{y}}^{\prime }=w\left(b,i\right)\times G_{y}$$
where w(b,i) (as Equation (Equation 3)) is an exponential function (where *i* represents the current row of the operator): *b* is an empirical value depending on the weight of the finger on the whole picture (after several trials, we chose to use b=50).
(3)w(b,i)=1−exp(−i/b)

For an image of size m×n, it is divided into two parts: the top and the bottom. This operator is used to convolve the two parts separately, then a gradient image in the horizontal direction of the finger vein image can be obtained. Next, the vertical midline of the gradient image is chosen as the baseline, and the maximum response value in the upper and lower parts of the search is found as the starting point of the upper and lower edges, noted as $\left(x_{u},y_{u}\right)$,$\left(x_{l},y_{l}\right)$.

#### 3.1.2. Finger Edge Search Rules Based on Finger Contour Imaging Properties

Generally, there is a very obvious difference between the finger area and the background area in the vein image, as shown by the high and low grayscale values (brighter in the finger area and darker in the background area). As shown in Figure 3, it is not difficult to observe the NMR image of the finger cross-section, which is nearly elliptical in shape, and the finger edge is relatively thin, so more infrared light can be penetrated, and higher gray value is displayed in the vein image. Therefore, the pixel value of the detected edge point is filled to 255, which ensures that the difference between the current point and the correct edge is minimized, and the difference between the current point and the background and the finger area is increased as much as possible to obtain the next edge point correctly (the principle is shown in Figure 4).

*F* is a finger vein image, F(x,y) is the pixel value of the point (x,y), correspondingly, *G* is a horizontal gradient map obtained from *F* by the improved horizontal Sobel operator we proposed. G(x,y) is the gradient value of the point (x,y) in the horizontal direction. The starting points $\left(x_{u},y_{u}\right),\left(x_{l},y_{l}\right)$ for tracking the finger edges have been obtained in the previous subsection, and the next step is to start searching for points on the edges (taking the upper right-hand edge as an example, and the same is true for the rest of the edges). Take the starting point $\left(x_{u},y_{u}\right)$ as the current position, diverge from this point to the right, find the next point, and put it into the set of coordinate points of edge points. The steps are as follows:
**Step 1** **Mark current point:** fill the pixel value of the current position of *F*, *G* to 255, respectively.
$$F\left(x_{u},y_{u}\right)=G\left(x_{u},y_{u}\right)=255$$**Step 2** **Update current point:** calculate the difference between the current point of *F*,*G* and each point of the lateral tri-neighbourhood, noted as $D_{F}^{1},D_{F}^{2},D_{F}^{3},D_{G}^{1},D_{G}^{2},D_{G}^{3}$, using Equation (Equation 4) to obtain the new coordinate point;
(4)$$\begin{array}{c} \left(x_{u},y_{u}\right)=argmin\left(\left(1-\theta \right)\times D_{F}^{i}+\theta \times D_{G}^{i}\right)\;\;\;i=1,2,3 \end{array}$$
where θ is the adjustable weight, generally, θ can be adjusted downwards when there is a large difference between the finger area and the background gray value;**Step 3** **Iterate over all columns:** repeat step 1 and step 2, starting with the baseline and traversing left and right through (Baseline,Baseline−1,…,2,1,0) and (Baseline,Baseline+1…,n−2,n−1), until each column of the finger vein image is completely traversed, so that a series of consecutive single-pixel edges can be obtained.

In the execution of step 2, errors are often generated because the upper or lower edge disappear (as shown in Figure 2d). Unlike the threshold or masking-based methods, the finger edge detection method we proposed in this paper is independent of the statistics of the original image pixels, so the image can be patched directly. For such a phenomenon, we only need to fill the first and penultimate rows of the image with the pixel values of 255.

After extracting the upper and lower edges simultaneously, we can obtain four sets of edge coordinates, noted as ${\mathrm{index}}_{\mathrm{lu}},{\mathrm{index}}_{\mathrm{ru}},{\mathrm{index}}_{\mathrm{ll}},{\mathrm{index}}_{\mathrm{rl}}$. Based on the obtained edge points, we seek out the minimum width of the internal tangent line of the finger edge, and use it as the initial horizontal reference line. The upper and lower division lines are recorded as ${\mathrm{up}}_{1}$ and $lower_{1}$, respectively.

### 3.2. Finger Image Orientation Correction

Correction of the deformed images ensures that the appropriate desired region can be extracted from each finger vein image for accurate feature extraction and matching, which greatly improves the efficiency and accuracy of the recognition system. We correct the images by the following two sub-steps:(1)Using least squares estimation, the finger midline was fitted based on the midpoints of the upper and lower edge points;
(5)$$\left\{\begin{array}{c} \left(x_{i},y_{i}\right)=\left[\left(x_{i}^{u},y_{i}^{u}\right)+\left(x_{i}^{l},y_{i}^{l}\right)\right]/2\\ \hat{K}=\frac{\sum_{i=1}^{n}x_{i}y_{i}-n\bar{x}\bar{y}}{\sum_{i=1}^{n}{x_{i}}^{2}-nx^{2}}\\ \;\hat{b}=\bar{y}-\hat{K}\bar{x}\\ \hat{x}=\frac{1}{n}\sum_{i=1}^{n}x_{i}\;\bar{y}=\frac{1}{n}\sum_{i=1}^{n}y_{i} \end{array}\right.$$
using the angle between the center line and the horizontal as the angle of rotation;
(6)$$\varphi =\left\{\begin{array}{c} -\arctan\left(\hat{K}\right),\;\hat{K}<0\\ \arctan\left(\hat{K}\right),\;\hat{K}\leq 0 \end{array}\right.$$(2)Obtain the rotated image using affine transformation;
(7)$$\left[\begin{array}{c} x^{\prime }\\ y^{\prime }\\ 1 \end{array}\right]=\left[\begin{array}{ccc} \cos\varphi & -\sin\varphi & y-x\cos\varphi +y\sin\varphi \\ \sin\varphi & \cos\varphi & y-x\sin\varphi +y\cos\varphi \\ 0 & 0 & 1 \end{array}\right]\;\left[\begin{array}{c} x\\ y\\ 1 \end{array}\right]$$

After the image has been affine transformed, the edges of the fingers will change accordingly. In this case, it is only necessary to translate the horizontal reference line obtained in Section 3.1. and the specific translation mode is related to the rotation angle. Equation (8) is as follows:(8)$$\left\{\begin{array}{c} up_{2}=up_{1}-2\times \varphi \\ lower_{2}=lower_{1}-2\times \varphi \end{array}\right.$$

### 3.3. Large Receptive Field Gradient Operators

ROI extraction of finger vein images requires rapid extraction of the same region from vein images collected from the same individual under different conditions. After the previous two steps, the upper and lower horizontal reference lines have been obtained. Next, a segmentation reference line needs to be found in the vertical direction. By understanding the nature of the finger’s structure, we know the gaps between the cartilage of the finger will be more transparent to infrared light under ideal conditions. In the digital vein image, the joint cavity is shown as two brighter areas. Qin et al. [16] found that the distance between two joints was 1.2–1.3 times the width of the finger at the distal joint by statistics. Thus, we selected the joint cavity as the vertical reference line for ROI extraction of finger vein.

In the study of existing joint cavity localization methods, we found that most of them locate ROI in a local scope. For human vision, the change of grayscale of joint cavity belongs to a wide range of gradation. The gray gradient area of the joint cavity generally accounts for a large part of the finger region. If the joint cavity is only observed in a small range, it is impossible to determine whether the joint cavity is located or not. Computer vision is similar to human vision in that the vertical orientation of the ROI is often inaccurate if only local area is searched at a time (As shown in Figure 5), This is the cause of error recognition in the positioning method based on column pixel accumulation value [8], and the sliding window positioning method [17]. Therefore, we propose a joint cavity localization method with large receptive field, and the search scope is about 20% of the finger region each time. It is described in the following steps:**Step 1:** 
for the vein images obtained after correction, the candidate regions are delineated using the horizontal reference lines obtained in Section 3.2;**Step 2:** 
the candidate region is reduced to a quarter of the original image using “double triple” interpolation to obtain $g^{{}^{\prime }}\left(x,y\right)$ of size ($m^{{}^{\prime }}\times n^{{}^{\prime }}$);**Step 3:** 
using the large receptive field gradient operator (size of (11 × 7) as shown in Figure 6) to extract the vertical gradient from finger vein images;**Step 4:** 
we chose to localize the distal joint cavity. The distal and proximal joint cavity positions is defined as $j_{r},j_{l}$, respectively, and the cumulative values in each of three columns on the gained gradient image are calculated. The minimum coordinate can be calculated by Equation (Equation 9). After restoring, we can obtain the distal joint cavity. The proximal joint cavity position is obtained by the finger width *w* at the distal joint. The detailed formula is as shown below:
(9)$$\left\{\begin{array}{c} S=\sum_{i}^{m^{\prime }}g^{\prime }\left(i,c:c+3\right),c=1,4,7,\ldots ,n^{\prime }-3\\ j_{r}=argmin\left(S\right)\times 3\times 4\\ j_{l}=j_{r}-1.25\times w \end{array}\right.$$

To ensure minimal loss of information about internal vein features of the finger area, we need to update the horizontal reference line again with the help of the vertical reference line. In this case, we limit the minimum internal tangent of the finger edge to the vertical reference line. Then, a new horizontal reference line is obtained by translating as described in Section 3.2.

### 3.4. ROI Extraction

The vertical and horizontal reference lines are obtained in Section 3.3. The image is corrected by affine transformation in Section 3.2. Then, an accurately located finger vein image is obtained for ROI extraction. The specific segmentation results are shown in Section 4.4.

## 4. Experiments

In order to validate the performance of the proposed ROI extraction method, we use four public available finger vein datasets as experimental data and design a total of three comparison experiments. The first experiment verifies the robustness and accuracy of the finger edge search method based on the finger contour imaging characteristics proposed in this paper. The second experiment verifies the robustness and accuracy of the large receptive field gradient operator to localize joint cavities. The third experiment compares the ROI extraction method proposed with representative methods in two different categories, and uses EER metrics to directly evaluate the performance of different methods. All comparison experiments in this article were conducted on a computer with a Lenovo Desktop Computer (China) with Intel Core i7-8700 and 8 GB RAM using PyCharm Community Edition (2020. x64).

### 4.1. Experimental Data

For testing the efficiency and robustness of the proposed method in this paper, we use four public finger vein image databases for experiments, including SDUMLA-HMT [18], a multi-modal feature database published by Shandong University, China; UTFVP [19], published by University of Twente, Netherlands; MMCBNU_6000 [20], published by Jeonbuk National University, Korea; FV-USM [21], published by Universiti Teknologi Malaysia. Notably, to ensure consistency with other datasets, we preprocessed all images in FV-USM, including: (a) rotating fingers to the horizontal direction; (b) cropping the part of fingertip and some background in the original image. The vein images of these datasets have different levels of complex background and noise. Table 1 shows the details of all datasets. Through the comparison on these datasets, the accuracy and robustness of the proposed method can be verified. Figure 7 shows typical images with different qualities (high, medium, and low) from different databases.

### 4.2. Compare Different Finger Region Segmentation Methods

In this experiment, we choose two representative finger region segmentation methods. In order to verify whether the proposed segmentation method has high accuracy and efficiency or not, we use the typical images introduced in Section 4.1 for comparison. The first one is the threshold-based ROI extraction method proposed by Kumar et al. [9] (on different datasets, we try to find the optimal binarization threshold); the second one is the coarse extraction method in the mask-based approach proposed by Lu et al. [12]. We give the performance of different methods on these images, and Figure 8 shows the results. We demonstrated the segmentation results of different methods for different quality images in different datasets. For each dataset, the first row shows the single-pixel contiguous edge points obtained by this method (for easy observation, we have labeled the points adjacent to the top and bottom of the edge as well); the second row shows the mask obtained using the method proposed in this paper (this method does not require a mask to split the finger area, the mask here is only used for visual comparison); the third row shows the mask obtained using the method proposed by Lu et al. [12]; the fourth row shows the mask obtained using the method proposed by Kuamr et al. [9]; the fifth row shows the original image.

From Figure 8, it is easy to see that the first and second methods do not perform well for the SDUMLA-HMT dataset with complex backgrounds. As the background images presented not only have similar pixel value distributions to the finger regions, but also produce gradients in the horizontal direction, this affects the threshold binarization and edge detection, respectively. The first method needs different thresholds for each dataset. If OTSU [22] is used uniformly for binarization, the results will be worse. By looking at the third column, we can easily see that the second method is not robust to boundary vanishing and blurring. However, these methods are properly solved after segmenting the finger region using the method proposed in this paper.

Table 2 shows the time consumption of different segmentation methods. Comparing with the two methods, the proposed method has the most robust and accurate segmentation effect. Combining the segmentation results (Figure 8) with the segmentation time consumption (Table 2), we can conclude that our proposed method has the best segmentation accuracy, and its time efficiency is much higher than the threshold-based method. For the problems such as boundary disappearance and misalignment, which cannot be solved by the coarse segmentation in the second method, our method can solve them with a simple treatment and the time efficiency can be comparable to that of coarse extraction in Lu’s method.

### 4.3. Comparison of Different Joint Cavity Localization Methods

In experiments, we will verify the stability and robustness of the proposed joint cavity localization method. In order to ensure the objectivity and justice of the experiments, we use three typical images of different quality in each dataset, which have a large impact on the localization of the joint cavity, compared with three commonly used methods of joint cavity positioning. The first method is a single row maximum cumulative pixel average to locate in Yang et al. [8]. The second method is a sliding window based locating method in Yang et al. [17]. The third method is a double sliding window based locating method in Qiu et al. [16]. Figure 9 shows the performance of the different methods on different typical images. The first column is the original images. The second column is the horizontal reference lines. The joint cavity localization is limited in the finger region, which is segmented by the horizontal reference line. The third column is the cumulative distributions of pixels per column in the finger region. The fourth column is the comparisons of the effectiveness of different methods in localizing the joint cavity of the finger (orange: maximum cumulative pixel average positioning in a single line; lime green: sliding window-based positioning method; green: double sliding window-based positioning method; red: the proposed method).

By observing the cumulative value of the column pixels in the finger region and the original image, we can find that the joint cavity is not always the brightest part of the finger region due to the light effect. As the sensitivity to light changes, the first and second methods based on cumulative pixel values are not accurate. When the end of the finger is detected, because the finger is thinner and more light is transmitted, the pixel distribution value in this region is higher, which has an impact on the positioning. The proposed method has obvious stability and robustness when compared with the first and second method. In addition, in contrast to the method using a double sliding window for positioning, our proposed method shows better performance against light interference. It is always positioned on the leftmost side of the distal joint cavity and has better stability.

### 4.4. The Process of the Proposed ROI Extraction Method

In this experiment, we use the typical images in Section 4.1 for ROI extraction again. Figure 10 demonstrates the results obtained for each step in the processing, it can intuitively display the robustness of the proposed method in this paper. Typical images present a large number of factors that affect matching performance, such as complex background noises, uneven illumination, boundary disappearance, etc. Figure 10 shows the ROI extraction process for four datasets images with various qualities. Each row represents the extraction process for one finger vein image. From the first column to the fifth column are: original image, finger edge image, final horizontal reference line positioning, vertical reference line positioning, and ROI.

As can be seen from Figure 10, when the background of finger vein images is relatively complex, our method can achieve accurate segmentation of the finger region. The proposed finger edge detection method is based on the stable structural properties of the finger, this characteristic is not easily altered by external influences during finger vein imaging. For some light interference problems in the finger region, such as uneven illumination and a large pixel distribution due to thin finger ends, the proposed large receptive field gradient operator can still achieve accurate localization of the distal joint cavity. Our proposed gradient operator with large receptive fields is searching for the finger joint cavity in a large region, it is very similar to human visual characteristics. For large range of grayscale gradients, large receptive field gradient operator has better robustness. We also applied two updates to the horizontal reference line, which ensures minimal loss of vein feature information within the ROI. Based on the above operation, we can obtain more stable ROI, and it can ensure the minimum variation between the same individual under different conditions.

### 4.5. Comparison of Matching Performance

Finally, the proposed method is compared with two existing representative methods on different datasets. The vein features were extracted using the maximum curvature method [23] and the repeated line tracking method [24]. Matching experiments were performed to evaluate the effectiveness of ROI extraction. The first method is the threshold-based ROI extraction method proposed by Kumar et al. [9] (Similarly, on different datasets, we try to find the optimal binarized threshold). The second method is the mask-based coarse extraction method proposed by Lu et al. [12]. In the first method, a fixed threshold is used to binarize the image, and the binary image is used to subtract the binary edge image generated by the Sobel operator, and the connected domain is analyzed to obtain a more complete finger region. The second method extends the Prewitt operator to obtain the finger edges to divide the finger region.

The maximum curvature method and repeated line tracking method are used to extract the pattern features. We use the template matching method for pattern matching. A single finger in each dataset is considered an independent individual. The template used for matching is an image fusion using all the ROIs of a single finger. The performance of the matching was assessed using the False Acceptance Rate (FAR) and the False Rejection Rate (FRR). Different thresholds give different FARs and FRRs, and the Equal Error Rate (EER) is the value when the FAR and FRR are equal. For the number of matches, we use the SDUMLA-FV (3816/6) dataset as an example. With the template matching method, each image only needs to be matched with the matching template, so genuine matching takes 3816 times, while imposter matching takes 2,423,160 (3816 × 635) times.

The ROC curves for the corresponding datasets are illustrated in Figure 11, and the corresponding quantitative matching results are shown in Table 3. The experimental results indicate that our proposed method is optimal in terms of accuracy, robustness, and efficiency. It should be noted that the matching performance relied heavily on multiple aspects, such as the image enhancement method and the setting of the feature extraction parameters. This paper presents mainly an effective and robust ROI extraction method for finger vein images, and does not cover the study of image enhancement and feature extraction. It is believed that better matching performance can be achieved through careful adjustment of image enhancement and feature extraction parameters.

## 5. Conclusions

This paper proposes a new ROI extraction method based on the characteristics of the original vein image. Firstly, according to the structural properties of the finger in its sub-elliptical shape, we propose a finger edge search method based on the imaging characteristics of the finger contour. It can achieve accurate detection of finger edges, obtain continuous single-pixel edges, and complete the division of finger areas and background areas. Secondly, based on the principle of the large scale grayscale gradient generated in the imaging of the joint cavity of finger, we propose a large receptive field gradient operator. It can accurately locate the leftmost of joint cavity in finger region and obtain a stable segmentation reference line in the same finger, and the horizontal reference line is corrected multiple times to ensure that feature loss is minimized. Finally, the accurate extraction of ROI is completed.

The experimental results illuminate that for the images with different qualities in each dataset, the proposed method can accurately extract ROI with good robustness. By comparing the effect with representative extraction methods in finger region segmentation and joint cavity localization, our method has the best robustness and accuracy. For all the validation data, the proposed method for implementing ROI extraction and matching has the lowest EERs. Without feature extraction parameter adjustment, the best EER reached 0.67%; compared with the other two methods, the performance is obviously improved.

In recent years, due to the global pandemic, it has been shown that finger vein identification technology needs to develop in the terms of being fast and contactless. Therefore, efficient and robust finger vein ROI extraction is bound to be the focus of future biometric research. Next, we will present a new ROI extraction method suitable for different acquisition devices and application scenarios, to provide high identification accuracy while ensuring contactless authentication.

## Figures and Tables

**Figure 1 sensors-21-04402-f001:**
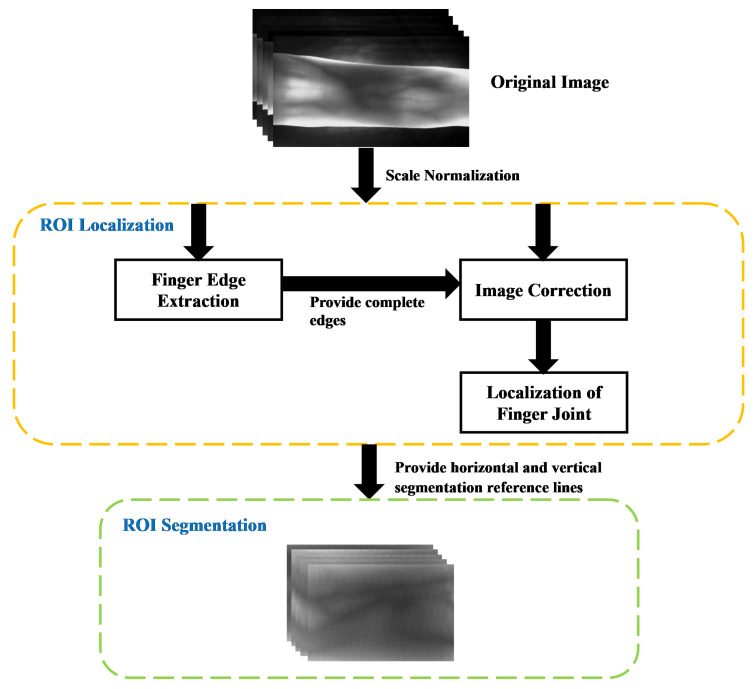
Block diagram illustration of the proposed method.

**Figure 2 sensors-21-04402-f002:**
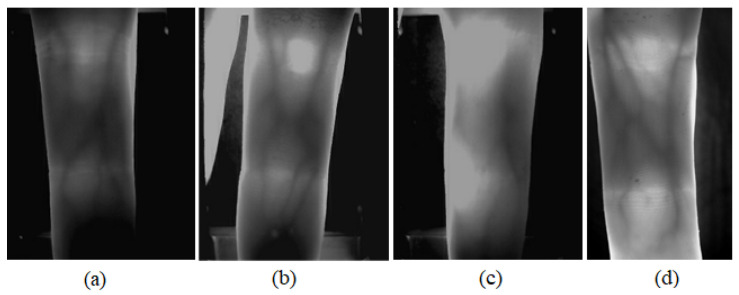
Typical images affecting edge extraction (**a**) Unclear borders (**b**) Wrong edge response (**c**) Unable to accurately binarize (**d**) Finger edges disappear.

**Figure 3 sensors-21-04402-f003:**
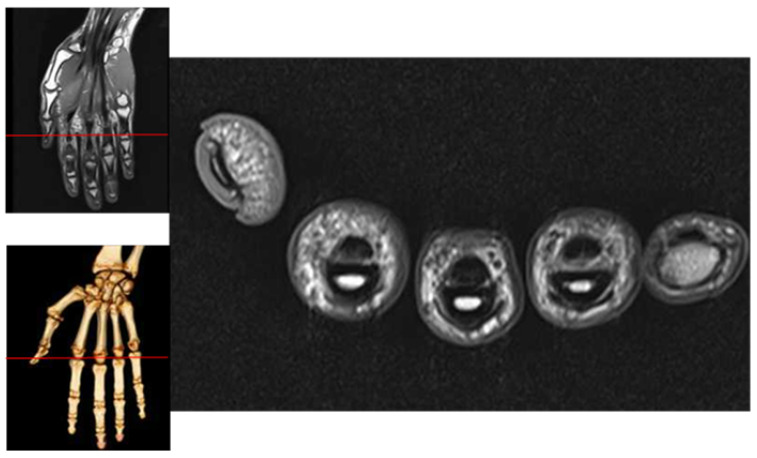
Cross-sectional MRI of the finger.

**Figure 4 sensors-21-04402-f004:**
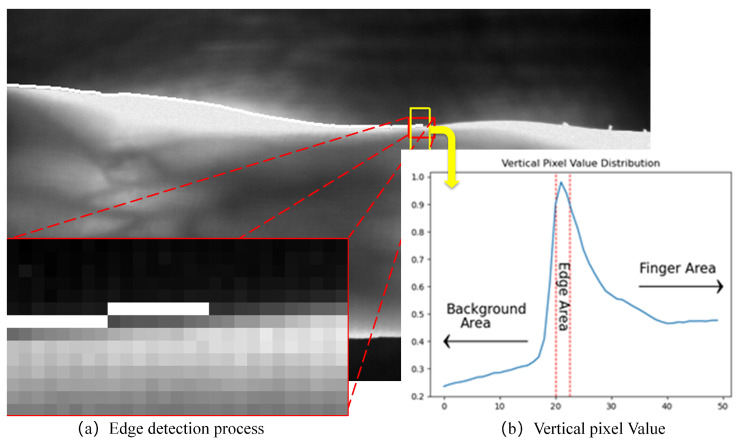
Edge detection process (**a**) Partial zoom during edge detection, (**b**) Column pixel accumulation in the area near the finger edge.

**Figure 5 sensors-21-04402-f005:**
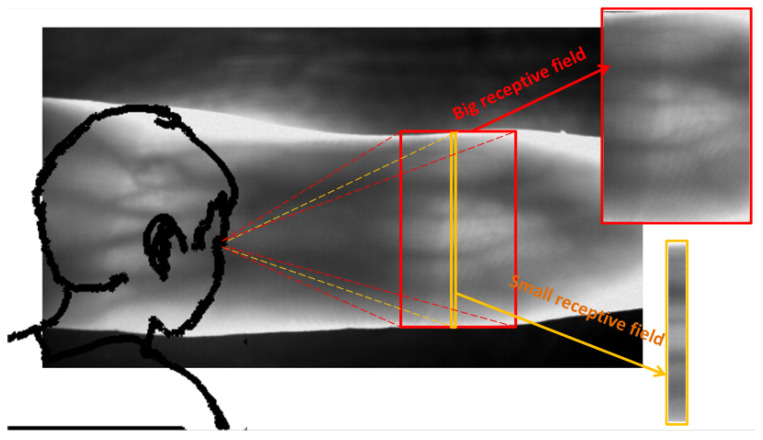
Human vision characteristics.

**Figure 6 sensors-21-04402-f006:**
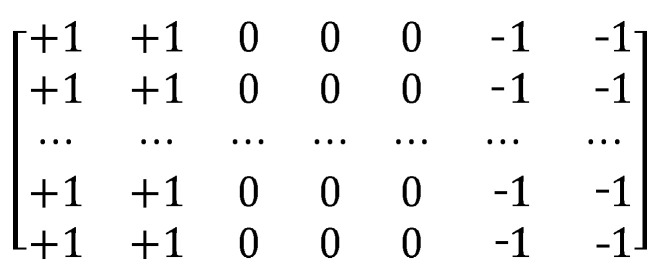
Large receptive field gradient operator.

**Figure 7 sensors-21-04402-f007:**
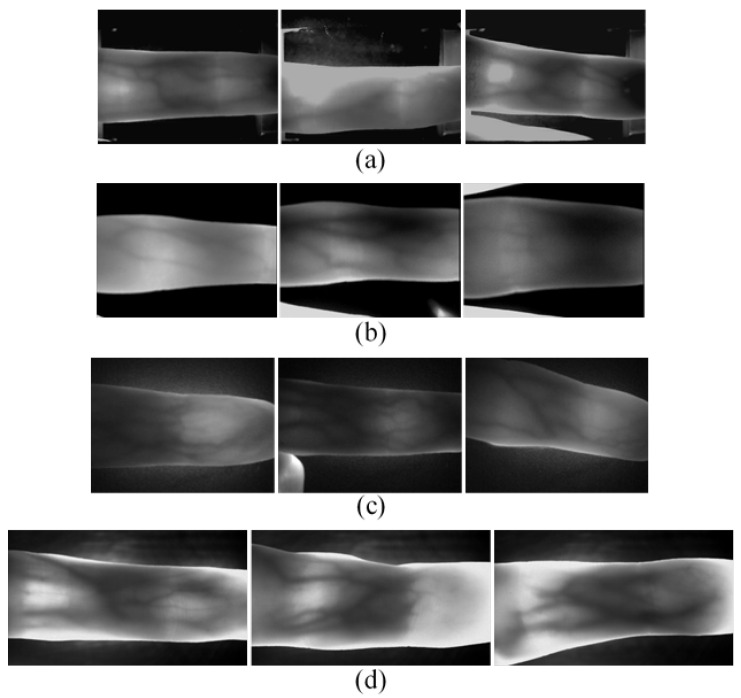
Typical images of three qualities in different databases. (**a**) SDUMLA-HMT (**b**) UTFVP (**c**) MMCBNU_6000 (**d**) FV-USM.

**Figure 8 sensors-21-04402-f008:**
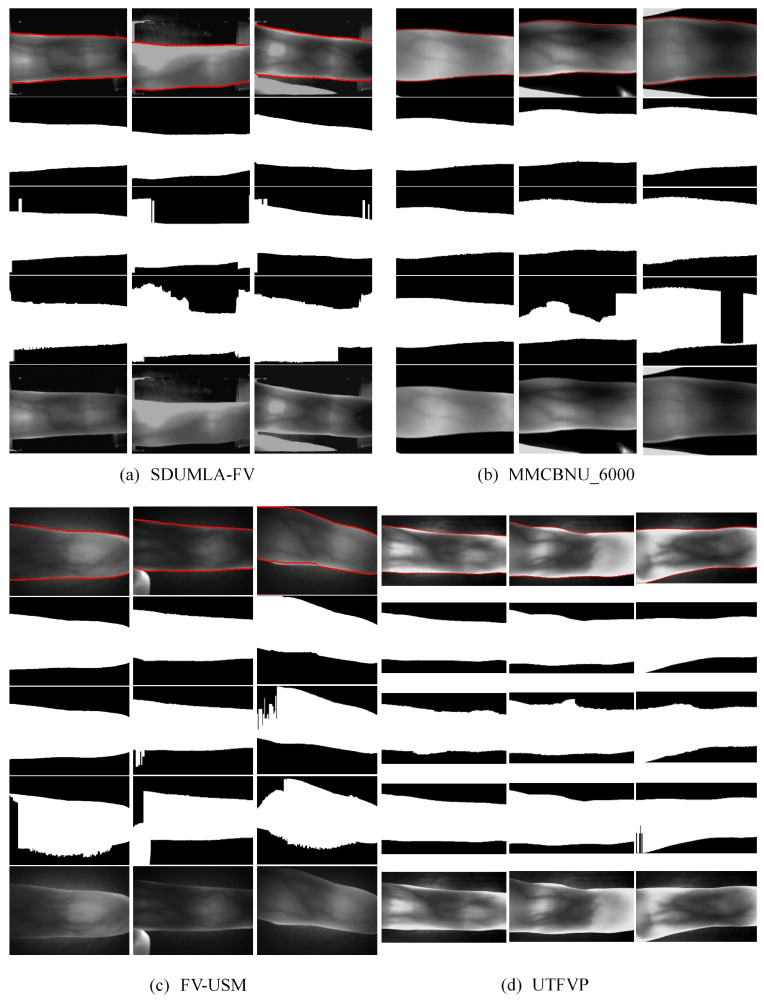
Comparisons of different finger region segmentation methods. For each dataset, first row: visualization of the results of the proposed method; second row: results of the proposed method; third row: results of the first method; fourth row: results of the second method; fifth row: the original images.

**Figure 9 sensors-21-04402-f009:**
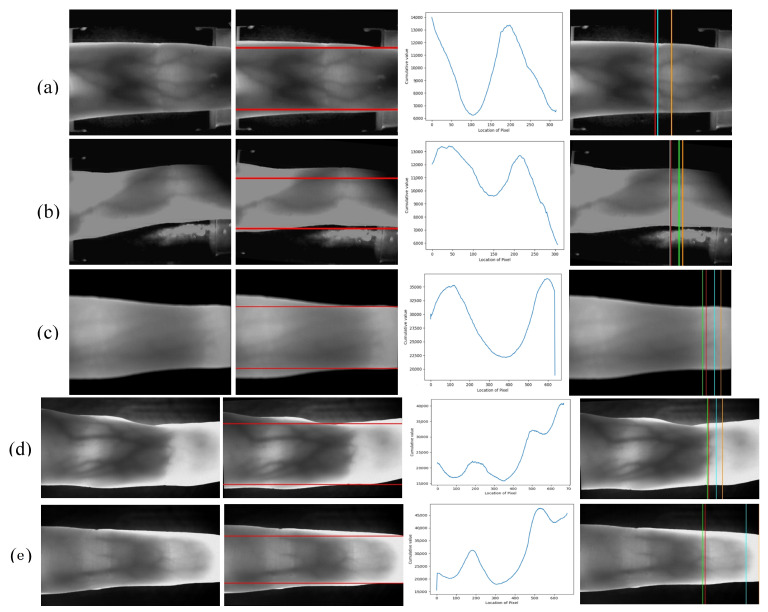
Comparison of different joint cavity localization methods. First column: the original images; second column: the horizontal reference lines; third column: Cumulative distribution trend for each column of pixels in the finger area; fourth column: comparisons of the effectiveness base on different methods in the joint cavity location of the finger.

**Figure 10 sensors-21-04402-f010:**
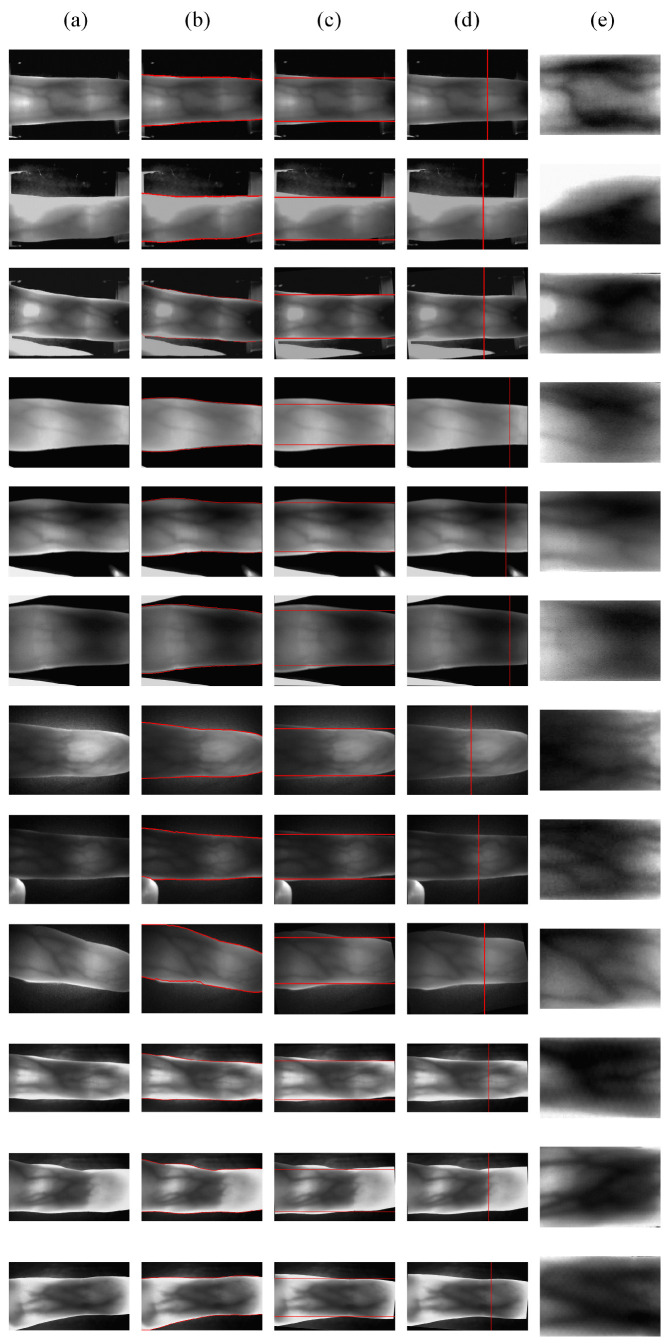
ROI extraction using the proposed method. (**a**) original images; (**b**) finger edges; (**c**) final horizontal reference lines positioning; (**d**) distal joints positioning; (**e**) extracted ROIs.

**Figure 11 sensors-21-04402-f011:**
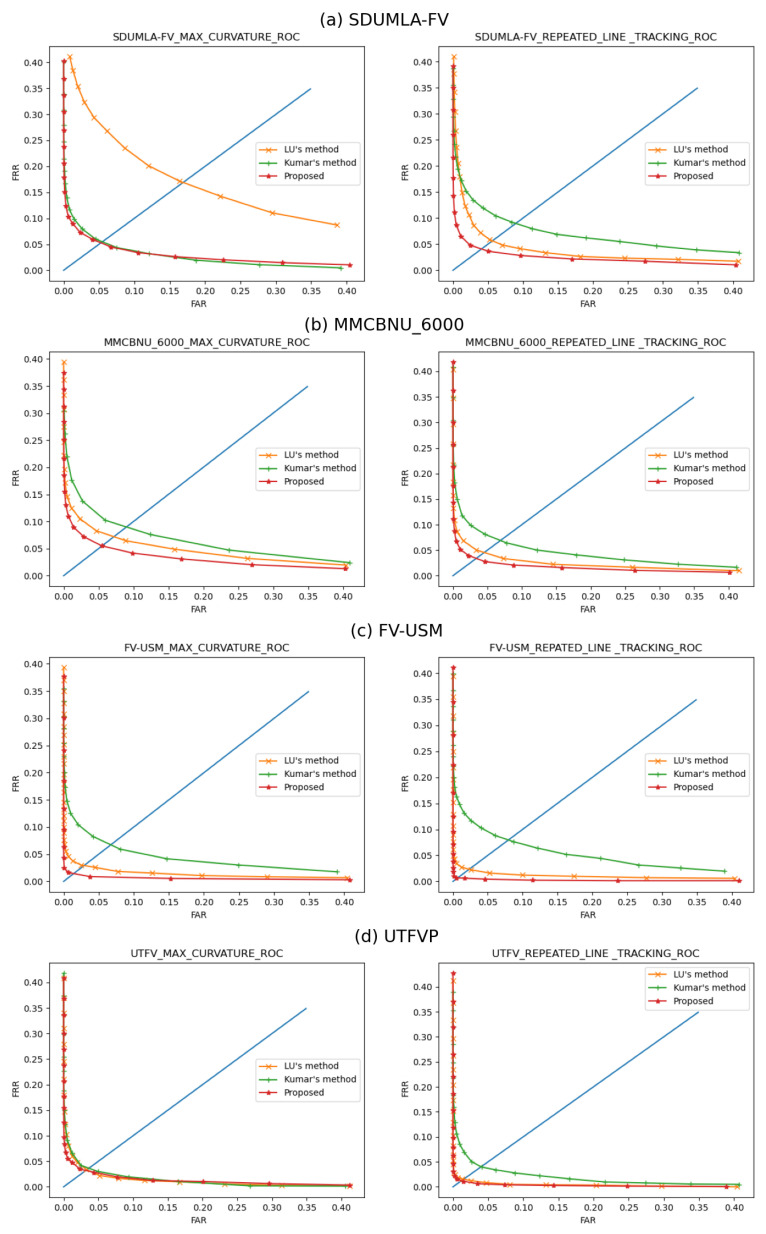
Comparisons of the ROC curves of different kinds of ROIs, obtained by using maximum curvature and repeated line tracking, respectively. (**a**) SDUMLA-FV, (**b**) MMCBNU_6000, (**c**) FV-USM, and (**d**) UTFVP.

**Table 1 sensors-21-04402-t001:** Details of the datasets used for the experiments.

Datasets	No. of Subjects	No. of Image	No. of Fingers for Each Subject	No. of Images for Each Subject	Image Size
SDUMLA-FV [18]	106	3816	6 *	6	320 × 240 pxl
UTFVP [19]	60	1440	6 *	4	672 × 380 pxl
MMCBNU_6000 [20]	100	6000	6 *	10	640 × 480 pxl
FV-USM [21]	123	5904	4 **	6	640 × 480 pxl

* both hands middle, index, ring. ** both hands middle, index.

**Table 2 sensors-21-04402-t002:** Comparisons of the time consumption using different finger area segmentation methods.

		SDUMLA-FV	MMCBNU_6000	FV-USM	UTFVP
Average time	Kumar’s Method	23.70 ms	60.10 ms	29.07 ms	49.99 ms
	Lu’s Method	1.74 ms	4.1 ms	1.97 ms	3.91 ms
	Proposed Method	1.69 ms	4.08 ms	2.12 ms	3.94 ms

**Table 3 sensors-21-04402-t003:** EERs calculation on different datasets using maximum curvature and repeated line tracking feature extraction methods (The best matches are marked in bold).

Feature	ROI	SDUMLA-FV	MMCBNU_6000	FV-USM	UTFVP
maximum curvature	Kumar’s Method	5.48%	12.17%	6.73%	3.64%
	Lu’s Method	16.90%	9.83%	2.92%	3.35%
	Proposed Method	**5.29%**	**5.49%**	**1.27%**	**3.19%**
Repeated line tracking	Kumar’s Method	8.98%	6.85%	7.97%	4.00%
	Lu’s Method	5.65%	4.56%	2.33%	1.53%
	Proposed Method	**3.96%**	**3.33%**	**0.67%**	**1.18%**

## Data Availability

The data presented in this study are openly available in [SDUMLA-FV, UTFVP, MMCBNU_6000, FV-USM], reference number [18,19,20,21].

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
