# Peer review of "A Novel ROI Extraction Method Based on the Characteristics of the Original Finger Vein Image"

_sensors, 2021, doi:10.3390/s21134402_

Round 1

Reviewer 1 Report

The global pandemic promotes the development of contactless identification. As the second generation of Biometrics identification technology, the study of non-contact finger vein recognition has certain practical significance. The ROI extraction method proposed in the article is different from the general edge extraction operator-based method and the threshold-based method. It provides a brand-new idea. The content
of the paper was very sufficient. With the popularity of finger vein recognition technology, the method authors proposed is likely to become a hot topic, and worthy of publication. The organization structure is more reasonable, and the methodology part is introduced in detail. In the experimental part, three experiments were designed on four databases with representative methods, and the experiments were very sufficient.
The detailed review of each part of the article is as follows:

(1) Abstract: The content is informative, complete and independent, and is a good summary of the full text;
(2) Introduction: Explains the purpose of the research and fully explains the research background;
(3) Related work: This part is very detailed and complete, and the related work review of ROI extraction is sufficient;
(4) Proposed method: The author discovered the characteristics of finger edges in finger vein imaging. The authors propose a new ROI extraction method for finger vein images, and introduce the proposed method in detail. Moreover, the existing methods and the characteristics of joint cavity imaging of finger vein are analyzed, and an effective solution for joint cavity positioning is given.
(5) Experiment: Three comparative experiments were designed and implemented, and the experiments were sufficiently verified to prove the effectiveness of the proposed method. The experimental analysis is also relatively sufficient.
(6) Conclusion: The conclusion of the article is too long and not concise enough. The important part should be selected for explanation.
This article mainly has the following problems to be solved:

minor:

(1) There are some words or sentences in the text that need to be further considered, or expressed in a more appropriate way;
(2) To ensure the completeness of the article, all the formulas in section 3.2 should be added with serial numbers;
(3) The image in the database FV-USM is different from the displayed image. It is necessary to explain in detail whether any preprocessing operation has been done;
(4) The best experimental results in the table need to be highlighted in bold;

major:
(1) The experimental analysis in section 4.5 is too simple and requires further detailed analysis;
(2) Conclusion: The conclusion of the article is too long and not concise enough. The important part should be selected for explanation.

Reviewer 2 Report

In this paper, ROI extraction method of finger vein image is proposed.

From the theoretical point of view, this method considered to be the combination and improvement of existing image processing techniques, however it is very practical, and shows good performance in experiments of authentication, with indirectly evaluating the correctness of extracted ROI by accuracy of authentication.

In Figure 8, ROI for each method is shown in white region, however it will be more clear if it is displayed on top of original image.

In line 347, "it is always positioned on leftmost side...", however it is not clear because the correct position of distal joint cavity is not shown in the figure.

In line 360, "only our method can...", however it can not be confirmed because results of other methods are not shown.

In 4.5, how did you detect the ROI of the template image for each method?
